

# Aerosol deposition to the boreal forest in the vicinity of the Alberta Oil Sands

Timothy Jiang[1*], Mark Gordon[1], Paul A. Makar[2], Ralf M. Staebler[2], Michael Wheeler[2]

[1]Earth and Space Science, York University, Toronto, M3J 1P3, Canada
[*]Now at School of Environmental Studies, Guelph University, Guelph, N1G 2W1, Canada
[2]Air Quality Research Department, Environment and Climate Change Canada, Toronto, M3H 5T4, Canada

*Correspondence to*: Mark Gordon (mgordon@yorku.ca)

**Abstract.** Measurements of size-resolved aerosol concentration and fluxes were made in a forest in the Athabasca Oil Sands Region (AOSR) of Alberta, Canada in August 2021 with the aim of investigating a) particle size distributions from different
sources, b) size-resolved particle deposition velocities, and c) the rate of vertical mixing in the canopy. Particle size distributions were attributed to different sources determined by wind direction. Background air from undeveloped forested areas air showed a peak number concentration for diameters near 70 nm while air mixed with upgrader smokestack plumes had higher number concentrations with peak number between diameters of 70 and 80 nm. Aerosols from the direction of open-pit mine faces showed number concentration peaks near 150 nm and volume distribution peaks near 250 nm (with
secondary peaks near 600 nm). Size-resolved deposition fluxes were calculated which show good agreement with previous measurements and a recent parameterization. There is a minimum deposition velocity of $v_d = 0.02$ cm s$^{-1}$ for particles of 80 nm diameter; however, there is a large amount of variation in the measurements and this value is not significantly different from zero in the 68% confidence interval. Finally, gradient measurements of PM1 demonstrated nighttime decoupling of air within and above the forest canopy, with median lag times at night of up to 40 min, and lag times between 2 and 5 min
during the day. PM1 fluxes determined using flux/gradient methods (with different diffusion parameterizations) underestimate the flux magnitude relative to eddy covariance flux measurements when averaged over the nearly 1-month measurement period. However, there is significant uncertainty in the averages determined using the flux/gradient method.

## 1 Introduction

Atmospheric aerosols are a strong influence on climate, affecting the radiation budget (both directly and indirectly) through radiative reflection, absorption, and influence on cloud formation. The net radiative effect is a large source of uncertainty in climate models (Boucher et al., 2013). Human exposure to particulate matter is linked to respiratory and cardiovascular disease and increased mortality, with no indication of a threshold value for health effects (Kappos et al., 2004). Forests comprise 9% of the land surface on Earth (Adams, 2012) and 40% of the land surface of Canada (NRCan, 2021) and they are
a net sink of aerosols due to dry deposition. Hence, modeling deposition to forests is key to correctly modeling atmospheric





aerosol concentrations. Aerosol deposition to forests also affects the health and growth of the forest (Matsuda, 2017). Hence, modeling deposition to forests is key to correctly modeling atmospheric aerosol concentrations.

The Athabasca Oil Sands Region (AOSR) in Alberta, Canada is the third largest oil deposit in the world. The activities associated with mining and bitumen processing in the region generate pollutants, greenhouse gases, and aerosols. The
aerosols include sulphates, black carbon, primary organic aerosols (POA), dust, and secondary organic aerosols (SOA) (Liggio et al., 2016) with SOA formation rates comparable to large North American cities. Aerosols thus originate in both direct emissions (primary) and in-situ reactions of gases (secondary).

Surface loss of aerosols through dry deposition is primarily dependant on aerosol size and vegetation type. Extensive reviews of dry deposition of aerosols can be found in Hicks et al. (2016), Saylor et al. (2019), Farmer et al. (2020), and Emerson et
al. (2020). The primary mechanism for the deposition of small particles < 100 nm in diameter is Brownian diffusion, which is more effective for smaller particles. The primary mechanism for the deposition of large particles > 300 nm in diameter is impaction and interception due to inertia. There is an intermediate size of particles for which both mechanisms are less effective, leading to the local minimum of deposition velocity of aerosols with respect to particle diameter, which is referred to in Hicks et al. (2016) as a "well" in deposition velocity of aerosols as a function of particle size. Emerson et al. (2020)
demonstrated that previous parameterizations overestimate deposition of the accumulation and Aitken mode particles and underestimate deposition in the coarse mode. The overestimation can be as large as an order of magnitude.

The minimum in deposition velocity has been observed in a coniferous forest in Southern Finland by Mammarella et al. (2011); however, these deposition velocities were not determined using size-resolved eddy covariance measurements. Instead, flux measurements were made using total number concentrations for particle diameters between 10 nm and 1 μm,
and then the size dependence is inferred using a particle deposition model. The Mammarella et al. study found local minima of aerosol deposition velocity at particle diameters 90 nm and 150 nm. The average deposition velocities in these bins were 1.8 mm/s and 1.9 mm/s, respectively. The location of the minimum "well" was recently demonstrated by size-resolved eddy-covariance aerosol flux measurements over a ponderosa pine forest by Emerson et al. (2020), who proposed a modification to the widely used Zhang et al. (2001) parameterization. The revised parameterization locates the minimum deposition
velocity near 70 nm in diameter (closer to the measured values), compared to the minimum located near 2 μm in diameter predicted by the Zhang et al. model.

The presence of the crown of a forest canopy leads to frequent decoupling between the sub-canopy space and the free atmosphere. Only occasionally (an average of 4 hours per day) does the sub-canopy air exchange energy and matter with the free atmosphere (Thomas and Foken, 2007), and mixing often does not occur for several hours at a time through the night.
The canopy is usually decoupled during calm conditions. Decoupling means that fluxes in and out of forests do not happen continuously but are discrete events (Foken, 2008). This is typically accounted for in models which include deposition by modifying the diffusion coefficient based on stability (e.g., Makar et al., 2017).

While forests are predominantly a sink for aerosols, forests can often be a source of aerosols to the atmosphere (e.g. Gordon et al., 2011; Pryor et al., 2008) either by adding biogenic mass to anthropogenic aerosols or by aggregation of organic matter.



Furthermore, the influence of mixing and coupling on deposition is often significant where stagnant air in the understory can act as a blocking layer between the canopy top and the surface (Schilperoort et al., 2020).

In spite of the abundance of aerosol deposition studies in forests, there has not yet been such a study in the AOSR. Several aircraft-based studies have characterized the composition and size of the oil sands aerosols in the region (Howell et al., 2014; Baibakov et al., 2021; Liggio et al., 2016) by flying through elevated plumes downwind of mines and upgrading facilities.

Howell et al. characterised the aerosols as a mix of freshly nucleated sulfates and nitrates, possible fly ash, and dust from dirt roads and mining operations, while Baibakov et al. found that the plumes were associated with elevated concentrations of sulfates and ammonium. All three studies demonstrated the formation of organic aerosol within 10's of km of the sources, and Liggio et al. demonstrated that SOA formation rates are comparable to megacities such as Mexico City or Paris.

The three primary goals of this study were to determine the sources of specific aerosol size distributions from oil sands

operations at ground level, to determine the size-resolved deposition rate of anthropogenic source aerosols into forests, and to study the effect the forest has on the vertical mixing of aerosols.

## 2 Methods

### 2.1 Site Location and Instrumentation

The YAJP tower was installed between Jul 2017 and Oct 2021 at 57.1225 N 111.4264 W. This work describes results from

three intensive field campaigns at the tower in 2017 (16 Jul – 1 Aug); 2018 (4 – 18 Jun); and 2021 (3 – 26 Aug). Size-resolved aerosols measurements were made during all three campaigns, while eddy covariance fluxes were measured only in the 2021 campaign. The results presented here focus on the Aug 2021 study only due to the availability of eddy covariance fluxes during the latter period.

The tower was mounted within the forest and is accessed through an unimproved road originally used for reflection

seismology. The closest paved road is the East Athabasca Highway which is a private road with generally light traffic approximately 650 m to the south of the tower. The site is surrounded by at least 10 km of boreal forest in all directions (Fig. 1) with oil sands open-pit mining, tailings, and processing facilities beyond that, predominately in the 135-270º and 305-45º sectors. The Athabasca River valley runs west of the site in a varying north-south direction, which is an influence on local wind direction. The village of Fort McKay is approximately 15 km to the NW of the site, and the town of Fort McMurray is

40 km south. Additionally, the Hammerstone limestone aggregate quarry is located 10 km NW of the tower.

The forest is mature jack pine (*Pinus banksiana*) and the ground is covered in reindeer moss (*Cladonia spp.*). The undergrowth in the area is limited to some sparsely distributed blueberry bushes. The ground is sandy and well drained. The forest's canopy height is approximated as 19 m (with the tallest trees in the area ranging from 16 m to 21 m in height). The one-sided leaf area index (LAI) was measured as 1.17 (based on Gap Light Analyzer software, Frazer et al., 1999) with a

stem density of approximately 320 trees ha[-1]. A generator was located 90 m from the tower, at a wind angle of 50º. During the entire 4-year duration of the project, less than 4% of the wind was from the 40-60º direction.



The ultra-high sensitivity aerosol spectrometer (UHSAS, Droplet Measurement Technology Inc.) measures particle number concentration in 100 size ranges between 55 nm and 1 μm. Particles are sized and counted with 1054 nm laser light scattered onto two magnifiers (one for small particles and one for large ones) collected by photodiode. It sizes up to 3000 individual

particles per second and size distributions for numbers higher than 3000 are scaled by total number count. Cai et al. (2008) have demonstrated that there is more than 50% loss for particles of size 55 - 60 nm, and an underestimation of the size of particles on the smaller end of its range. We restrict our analysis to the 60 nm to 1 μm range. Size distributions were sampled at a 1 Hz frequency. The UHSAS was installed at the base of the tower and sampled from a height of 29 m through a 32 m length of 3/8" ID static-dissipative tubing. The measured flow rate of 15 L min$^{-1}$ results in a delay time of 9 s.

A 3D sonic anemometer (Type A, Applied Technology Inc.) was co-mounted with the UHSAS inlet at 29 m. The anemometer faced approximately south (169º) and was mounted 0.7 m from the tower structure (which is an open triangular cross-section with 0.4 m sides). For the 2017 field study, a second anemometer was mounted at a height of 9 m within the canopy. The anemometers sampled at a frequency of 10 Hz.

Two particle counters (DustTrak DRX 8533, TSI) were mounted at ground level (2 m) and at a height of approximately 20

m. The DustTrak particle counters sample 2-min averages of total particle mass for an approximate range of 0.1 to 15 μm (size resolved into PM1, PM2.5, PM4, and PM10 size ranges). Here, we only used the PM1 (0.1 to 1μm) size range. The instruments were set to auto-zero every 15 min. Yun et al. (2015) found that the DustTraks require a correction factor of 0.29 for PM2.5. The DustTrak measurements were concurrent with the UHSAS particle concentration sampled at 29 m. This allowed a comparison of PM1 at heights of 20 m (DustTrak) and 29 m (UHSAS), both above the canopy height of $h_c = 19$

m. Assuming little variation in concentration between these two heights, and an average aerosol density of 1500 kg m$^{-3}$, the measurements suggest a correction factor of 0.5 ( $R^2 = 0.97$). This correction factor is applied to the DustTrak measurements, although the uncertainty due to the height difference is discussed.

A $CO_2$ and $H_2O$ gas analyzer (LI-7500, Licor Inc) mounted at a height of 29 m near the anemometer was used to measure latent heat flux to correct for density fluctuation (Webb et al., 1980).

**2.2 Flux Calculation**

Eddy covariance could not be calculated for the 2017 and 2018 studies due to an excessive residence time in the 32 m sampling line. This was corrected for in the 2021 study by reducing the inlet tubing diameter and hence increasing the flow rate. Fluxes were calculated in 30-min periods in each size bin following the eddy covariance method. The coordinate system was rotated around the $z$ and $y$ axes to give $\overline{v} = \overline{w} = 0$ (overbar denotes 30-min average) following Wilczak et al., 2001. To

remove spikes in anemometer data caused by electronic noise and processing errors, three passes removed all high-frequency data within each 30-min period more than 5 standard deviations from the mean. This removed less than 0.05% of the data. The generator used to power the instrumentation was place downwind of the prevailing wind direction. All data from the 40-60º direction were removed to avoid contamination by the generator exhaust, resulting in a removal of approximately 5% of the measurements during the 2021 measurement period. To remove conditions with low turbulent mixing, which are





considered unreliable for eddy covariance, periods with friction velocity of $u_* < 0.2$ m s$^{-1}$ were removed, resulting in a further removal of 17% of the measurements.

Fluxes were corrected for density fluctuations due to water vapour following Webb et al. (1980). No corrections were made for fluctuations in density due to heat flux, as fluctuations of heat are assumed to be dissipated in the 32-m inlet tube (Rannik et al., 1997). The average density correction was less than 6%. Finally, fluxes were also corrected for the attenuation of the

signal carried by frequencies >1 Hz due to the sampling frequency of the instrument, following Horst (1997), which resulted in an average increase of 16%.

Due to the proximity of the aerosol sources, the aerosol measurements at the site vary considerably as changes in wind direction shift the plume within 30-min periods. We apply linear detrending of each $N(t)$ 30-min time series to account for variation in aerosol measurements through the 30-min period.

**3 Results and Discussion**

**3.1 Source Characterization**

The 30-min particle number concentration measurements are shown with wind direction in Figure 2. To focus on consistent winds and to avoid recirculating wind patterns, all 30-min observations with greater than 20° difference relative to either the preceding or following measurement are removed, resulting in a removal of approximately 30% of the data. Based on these

results, we identify seven sets of measurement of interest based on wind direction and concentration. These sets are then used to investigate differences in the size-resolved number and volume distributions of sub-micron aerosols and to attempt to attribute these differences to anthropogenic emission sources. While we attempt to correlate these sets in direction-concentration space with the location of sources in the region, it is recognized that these sets are arbitrarily defined and there is likely some overlap of multiple sources in the size-resolved number and volume distributions associated with each set.

Below we follow a 360° circuit around the YAJP site comparing the measurements shown in Figure 2 with the site locations shown in Figure 1 (using Google Earth images from August 2021 for more precise identification of potential source types and locations). While back-trajectory models such as Hysplit could offer a better indication of the source location than the wind direction measured at the site, it has been demonstrated that the model wind fields do not have sufficient resolution to resolve local topography in this region such as the river valley (Yousif et al., 2022).

Measurements from the wind direction between approximately 10° and 40° show a mixture of low (~0.1×10$^9$ m$^{-3}$) and high (~2×10$^9$ m$^{-3}$) aerosol concentrations. The Shell Jackpine site (Fig. 1) has an active mine face more than 10 km from the YAJP site between 10° and 25°. Here we isolate the high-concentration (1.5×10$^9$ – 2.5×10$^9$ m$^{-3}$) measurements for wind directions between 5° and 40° (Fig. 2, set i) and designate these as anthropogenic emissions from the Shell Jackpine site. Although there are lower concentration measurements (<1.5×10$^9$ m$^{-3}$) from the same wind direction, the measurements

within set i show a more consistent mode diameter.



Measurements from between 60° and 135° are sparse (since this is not a prevailing wind direction) and consistently low concentration. This region (Fig. 2, set ii) is purely forest (Fig. 1) except for the Suncor Firebag site, which is in-situ mining more than 30 km away and would not be expected to have significant PM emissions.

Measurements from near 180° are always above background levels ($> 0.5\times10^9\,\mathrm{m^{-3}}$) and span a range of concentrations up to 165 $<2.5\times10^9\,\mathrm{m^{-3}}$ with a single outlier near $4\times10^9\,\mathrm{m^{-3}}$. Although the measurements from this region do not show distinct sets in this direction-concentration space, it is noted that Suncor open-pit mining is upwind of the site between directions of 150° and 180° and the Suncor upgrading facility (and the main smokestack) is at an upwind direction of 192° relative to the YAJP site. Additionally, Fort McMurray is approximately 45 km south of the site and Highway 63 runs north of Fort McMurray along the river valley. Here we recognize that the north-south valley system will likely affect wind patterns and could turn 170 prevailing SW winds into southerly directions. Hence, a 180° wind direction measurement at the YAJP site may correspond to a source direction of >180°. Despite this, we divide these southerly wind direction measurements into two sets: relatively high concentrations ($> 1.2\times10^9\,\mathrm{m^{-3}}$) between 150° and 200° (Fig. 2, set iii) and relatively low concentrations ($< 1.2\times10^9\,\mathrm{m^{-3}}$) between 150° and 220° (Fig. 2, set iv).

$SO_2$ measurements at YAJP shown in Gordon et al. (submitted with this manuscript) demonstrate elevated $SO_2$ in the 160° to 175 250° range, which is a subset of the elevated aerosol measurements in the 150° to 280° range shown in Fig. 2. Since $SO_2$ is primarily emitted from smokestacks (Zhang et al., 2018), this implies that sets iii and iv contain a mixture of both smokestack and open-pit mining emissions and that plume emissions from smokestacks cannot be completely isolated from open-pit sources for Suncor and Syncrude.

The Syncrude upgrading facility (and the main smokestack) is at an upwind direction of 234° relative to the YAJP site. 180 Active open-pit mines range from 235° to 270°. In the 230° to 280° range (Fig. 2) there are some near-background concentrations ($<0.6\times10^9\,\mathrm{m^{-3}}$), somewhat separate from relatively high concentrations spanning $0.7\times10^9\,\mathrm{m^{-3}}$ to $2\times10^9\,\mathrm{m^{-3}}$. To attempt to differentiate between stack emissions near 235° and open-pit mining sources, we divide these relatively high concentrations into two sets: from 230° to 255° (Fig. 2, set v) and 255° to 280° (Fig. 2, set vi) noting again that channeling of the flow along the river valley may result in a measured wind direction that does not correspond exactly to source direction. 185 Similar to sets iii and iv, sets v and vi likely contain a mixture of smokestack and open-pit mining emissions.

Finally (completing the 360° circuit around the YAJP site), the region between 275° and 305° is mostly forest and includes the town of Fort McKay at a distance of 15 km from the YAJP site, while the region between 305° and 360° includes CNRL (~30km), the Hammerstone limestone aggregate quarry (10 km at 315°), the Shell Muskeg River (15 km at 330°) and Syncrude Aurora (20 km at 345°). Despite these multiple sources within this sector, the number concentration for wind 190 directions > 275° is relatively low (except for a few outliers near $1.7\times10^9\,\mathrm{m^{-3}}$). Here we define a set of measurements for wind directions between 295° and 335° and concentrations up to $1\times10^9\,\mathrm{m^{-3}}$ (Fig. 2 set vii).

Particles size distributions (PSDs) for the sets defined above are shown in Figure 3. Set i, which is likely from the Shell open pit mining, shows the greatest number and volume with a peak number concentration for diameters near 150 nm and a peak volume concentration for diameters near 300 nm (with a smaller secondary volume peak near 600 nm). When the number





PSD for set i is normalized, it is similar in shape to set vi with the exception of a small peak near 70 nm -in set vi. Set vi is likely associated with Syncrude open-pit mining and also peaks at a diameter near 150 nm. The volume PSDs of sets i and vi (presumably from Shell and Syncrude open-pit mining, respectively) are similar but set vi does not show a strong secondary peak for large (~600 nm) particles. This may be due to the relative proximity of the Shell mines (~10 km) versus the Syncrude mines (~15 km) as these larger particles may have deposited over the longer upwind fetch.

The number and volume PSDs of set vii are very similar to those of set ii, suggesting that the winds from both the east and NW regions bring background biogenic forest aerosols to the YAJP site. Sets iii and iv, which are both from the direction of Suncor, are nearly identical when normalized, suggesting that the lower concentrations (set iv) are from the same source, but diluted with background air. These PSDs show peak number concentration for diameters between 70 and 80 nm, suggesting newly formed particles from upgrader stack emissions (Zhang et al., 2018). The normalized number PSDs show two distinct

PSD shapes: those dominated by new, smaller (~75 nm) particles or biogenic background particles (sets ii, iii, v) or those dominated by larger (~150 nm) particles (sets i, iv) associated with open-pit mining. Sets v and vi appear to be a mix of these two PSDs, possibly owing to a mix of Syncrude stack emissions with Syncrude open-pit mining emissions.

It is perhaps surprising that the normalized number PSDs associated with biogenic background air (sets ii, vii) are very similar to the normalized number PSDs associated with stack emissions (sets iii, iv), despite the stack emissions having

number concentrations up to a factor of 20 higher than the background emissions. However, the normalized volume PSDs associated with background air are very different, with peak volume concentrations for diameters ranging from 200 to 400 nm (compared to 250 nm for all other PSDs).

Howell et al. (2014) measured PSDs for number and volume from an aircraft within a plume 10 km and 182 km downwind of the plume source. The particle diameter corresponding to the peak number density was approximately 15 nm at 10 km

downwind and close to 60 nm at 182 km downwind. A smaller secondary peak similarly shifted from near 100 nm to 150 nm. Particle volume peaked between 100 and 200 nm at both 10 km and 182 km downwind of the source with a secondary peak at 182 km downwind near 70 nm. At the ground level approximately 15 km from the source, we observe a number peak near 75 nm (smaller than the Howell et al. aircraft observations) and a volume peak close to 300 nm (larger than the Howell et al. observations).

Baibakov et al. (2021) also measured PSD from an aircraft for two distinct plumes downwind of both the Syncrude and Suncor upgraders. One plume had significantly higher total number concentration by a factor of ~2 and a peak volume near 600 nm (similar to the smaller secondary peak seen in out measurements). The lower number concentration plume had a peak volume near a diameter of 240 nm, with a volumetric PSD similar to the background (out-of-plume) PSD. Hence YAJP surface-based results demonstrate measured plume and background PSDs with a range of distributions that show significant

difference from PSDs measured from aircraft in the region.



### 3.2 Flux Spectra

The flux spectra allow us to test that the instrument frequency is fast enough for flux covariance and that there is no substantial dispersion or diffusion in the inlet lines. Specifically, evidence of an inertial subrange at high frequencies (a variation of covariance with frequency that follows a −7/3 power-law) demonstrates that the eddy-covariance measurement has captured the contributions of the energy-containing eddies (e.g., Foken, 2008). Normalized co-variance spectra of 30-min $\overline{w'N'}$ fluxes are shown in Figure 4, where $N$ is the aerosol number concentration for sizes between 60 nm and 1 µm. These spectra were randomly selected from the 15–26 Aug period (the second half of the study when measurements were more consistent through the day). The spectra are separated into two groups according to the slope of the inertial subrange (here defined as $f > 0.1$ Hz). The normalized co-spectra multiplied by frequency ($fS_{wN}$) should vary following a −4/3 power-law in the inertial subrange (Kaimal and Finnigan, 1994). The spectra shown in Fig. 4a are close to this ideal, although the slope can be near –1 in some cases, possibly due to instrument noise. Heat and $CO_2$ fluxes measured at the site (not shown) demonstrate a power-law slope of −4/3 in the inertial subrange for $f > 0.1$ Hz. For the sample periods shown in Fig. 4b, no inertial subrange is seen and the contribution from higher frequencies is comparable to the lower frequency contributions. This is likely due to the flux signal being small relative to the noise caused by the instrument or diffusion in the sampling tube.

The presence of the inertial subrange is related to the strength of the flux. Using a least-squares fit to the spectra for $f > 0.1$ Hz, an inertial sub-range slope ($S$) is calculated for each spectrum. The average flux magnitude for flux spectra with slopes $S < –2/3$ (69 30-min values) is $\left|\overline{w'N'}\right| = 1.44 \times 10^7$ m$^{-2}$ s$^{-1}$, while the average flux magnitude for slopes $S > –2/3$ (596 30-min values) is $\left|\overline{w'N'}\right| = 5.2 \times 10^6$ m$^{-2}$ s$^{-1}$ (a factor of 2.8 smaller). Similarly, the average number density is higher for the steeper slopes ($\overline{N} = 1.2 \times 10^9$ m$^{-3}$ for $S < −2/3$ and $\overline{N} = 6.7 \times 10^8$ m$^{-3}$ for $S > −2/3$). Approximately 83% of the strong spectra ($S < −2/3$) are measured during the daytime (between 07:00 and 17:00) when there is greater flux of aerosols into the canopy (due to stronger turbulence). This demonstrates that the presence of the inertial subrange is associated with a strong flux measurement and lower flux measurements lead to a flattening of the inertial subrange. Although the lack of an inertial subrange may indicate a significant noise to signal ratio, these weak spectra (which are associated with lower flux and concentration values) are included in our analysis since removal of these data would introduce a strong daytime bias in the results.

### 3.3 Deposition Velocity

Figure 5 shows the size resolved deposition velocities, including average values with standard error (equivalent to 68% confidence interval), median and 25th and 75th percentile values. The size resolution is reduced to every third size bin to improve clarity and reduce noise (errors are averaged in quadrature). The substantial variation in the measurements is demonstrated by the 25th and 75th percentiles which span ~2 cm s$^{-1}$ in the 60 to 215 nm size range. This demonstrates substantial exchange of aerosols in both directions (into and out of the canopy) with a net deposition that is a small fraction



of that variation. Within the 65 to 130 nm size range $v_d$ values are not significantly different from zero in the 68% confidence interval (C.I.). For the 130 to 250 nm size range $v_d$ values are significantly different from zero in the 68% C.I.

but would not be significantly different from zero in the 95% C.I. (2 standard errors). For sizes greater than 300 nm, there is substantial variation in the values of $v_d$ between neighbouring size bins and a consistent variation of $v_d$ with size is not seen. The average values are compared to previously published Emerson et al. (2020) measurements and parameterization in Figure 6. Emerson et al. develop the parameterization shown in Figure 6 based on 126 measurement points as summarized by Hicks et al. (2016), Saylor et al. (2019), and Farmer et al. (2020). The factor of 5 parameterization bounding range used

by Emerson et al. (2020) and shown in Figure 6 encloses 110 (87%) of these previously published data points. Emerson et al. (2020) present recent size-resolved deposition velocity measurements (not included in the review papers) made using eddy covariance with a UHSAS instrument in a ponderosa pine forest. Theory predicts a minimum value in $v_d$, which Hicks et al. refers to as a "well" in the size distribution of $v_d$. The parameterization proposed by Emerson et al. (shown in Fig. 6) predicts a minimum value of $v_d = 0.12$ cm s$^{-1}$ near a particle diameter of 62 nm.

The measurements of this study (shown as black squares in Figure 6) show good agreement within the range of previously reported values of $v_d$ for the measured size range of 60 nm to 1 μm, particularly with the Emerson et al. results over the same size range. A local minimum of $v_d = 0.02$ cm s$^{-1}$ is observed at 80 nm. Based on the standard error of the measurements, this minimum deposition velocity, and many of the measurements for similar particle sizes, are not significantly different from zero at the 68% C.I. (error bars extending below 0.01 cm s$^{-1}$ in Fig. 6). We also note that for sizes

greater than 360 nm these are strictly minimum positive values (not minimum values), since many size bins in this range (where number concentration is very low) report negative values of $v_d$ (Fig. 5) which cannot be plotted on a logarithmic scale. The minimum measured $v_d = 0.02$ cm s$^{-1}$ is much less than the modeled minimum value of $v_d = 0.13$ cm s$^{-1}$ from the Emerson $v_d$ formulation and is outside the parameterization bounding box, but this value is closer to the minimum measured value of Emerson et al. of $v_d = 0.05$ cm s$^{-1}$ for a particle diameter near 86 nm.

**3.4 Canopy Decoupling and Gradients**

Time series measurements of particle mass concentration (PM1) made with the DustTraks at heights of 2 m and 20 m demonstrate a lag between the 20-m and 2-m measurements which is more pronounced during the night. This is indicative of the decoupling between the canopy-top and sub-canopy where changes in concentration due to advection above the canopy take longer to reach the sub-canopy during stable conditions (Thomas and Foken, 2007). Time lags on the order of 2 hours

have been observed during the night for aerosols in other forests (Gordon et al., 2011; Whitehead et al., 2010). Investigation of this effect helps to improve deposition modeling by including the decoupling effect through stability parameterizations.

The lag is determined at the YAJP site as the time a change in concentration at 20 m takes to appear in the 2-m measurement time series. Figure 7 shows the lag binned by hour of day. The jack-pine boreal forest here is much less dense forest than the mixed forest of Gordon et al. (2011) or the tropical rainforest of Whitehead et al. (2010). Hence, the time lags are smaller,

with peak median values near 40 min between 03:00 and 05:00. Through the afternoon, the median time lags range from 2 to 5 minutes.

The average aerosol total PM1 mass flux (integrated over the size range of 60 nm to 1 μm) determined by eddy covariance with UHSAS measurements assuming a particle density of 1200 kg m$^{-3}$ (following Emerson et al., 2020) is $\overline{w'C'} = -10.8$ ng m$^{-2}$ s$^{-1}$ (with a 68% C.I. of 5.3 ng m$^{-2}$ s$^{-1}$). By comparison, an average total mass flux (positive upwards) can be calculated from a 2-point flux/gradient relationship as

$$F = -K \frac{\Delta C}{\Delta z}, \tag{1}$$

where $\Delta C$ is the concentration difference between two heights separated by $\Delta z$, and $K$ is the average diffusion coefficient between the two heights.

One approach is to parameterize $K$ based on a measured 2-point wind gradient and momentum flux (You et al., 2021 and Gordon et al., submitted with this manuscript). Here Prandtl's mixing length model is adjusted for stability following Garratt (1996) to give

$$K = \frac{\kappa \, z_m}{Sc} \frac{u_*}{\phi}, \tag{2}$$

where $\kappa = 0.4$ is the von-Karman constant, $z_m$ is a representative flux measurement height, $Sc = 0.8$ is the turbulent Schmidt number (ratio of momentum diffusion to trace gas diffusion), $u_*$ is the friction velocity, and $\phi$ is a stability parameter. Measurements at this forest tower site outlined in Gordon et al. (submitted with this manuscript) demonstrate good agreement ($R^2 = 0.83$) between $K$ determined by the measured flux and the gradient with a representative height of $z_m = 11$ m, which is roughly the middle of the canopy height ($h_c = 19$ m).

Another approach specific to forest canopies uses a vertically varying diffusion coefficient following Raupach (1988) as $K(z) = \sigma_w^2 \, T_L$, where the Lagrangian timescale $T_L = 0.3 \, h_c \, \sigma_w^2 \, / \, u_*$ and $\sigma_w^2$ is the vertical velocity variance, which varies with height. If it is assumed that $\sigma_w$ varies linearly from zero at the surface up to a value of $\sigma_w(z_m)$ at height $z_m$, then the average $K$ over the height of the canopy is

$$K = 0.1 \, h_c \frac{\sigma_w^2(z_m)}{u_*}. \tag{3}$$

Makar et al. (2017) also propose a vertically varying $K(z)$ parameterization specific to forest canopies based on measurements from a number of studies (references therein). Here we vertically average this parameterization through the canopy height to give

$$K = u_* f\left(\frac{h_c}{L}\right), \tag{4}$$

where $f(h_c/L)$ is a function based on the Obukhov length ($L$). The function is nearly linear between $f = 2.35$ m at $h_c/L = -0.1$ and $f = 0.38$ m for $h_c/L = 0.9$ and is constant outside those limits.



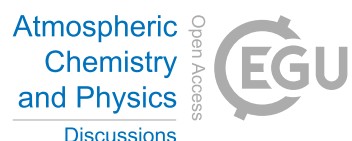

Table 1 compares the flux calculated using the flux/gradient method using these three diffusion parameterizations (Eq. 2-4) against the average PM1 mass flux calculated with eddy covariance. The concentration difference $\Delta C$ is calculated using the PM1 mass concentration measured by the DustTraks (integrated from 0.1 to 1 μm) with the 0.5 correction factor discussed in Section 2.1. The values range are constantly smaller in magnitude than the eddy covariance flux by a factor ranging from 2 (Eq. 2) to 9 (Eq. 4). Due to the relatively small differences between the upper and lower concentrations, there is a large amount of noise in the gradient ($dC/dz$) and only two of the three flux/gradient averages are significantly different from zero at the 68% C.I. Regardless, this implies that the diffusion coefficients are underestimated. This also demonstrates the degree of uncertainty involved in parameterizing a diffusion coefficient through the vertical extent of the canopy since the 68% C.I. are close in magnitude to (or greater than) the average flux values.

## 4 Conclusions

YAJP surface-based results demonstrated measured plume and background PSDs with a range of distributions that show significant difference from PSDs measured from aircraft in the region. Measurements suggest that larger (> 500 nm diameter) particles are from open-pit mining to the north and south of the tower and smaller (<100 nm diameter) particles are from stack sources from the west and south-west directions. Background air from undeveloped forested areas showed a peak number concentration for diameters near 70 nm. Air containing plumes mixed with upgrader smokestack plumes had higher number concentrations with peak number between diameters of 70 and 80 nm. Aerosols from the direction of open-pit mine faces showed number concentration peaks near 150 nm and volume distribution peaks near 250 nm (with secondary peaks near 600 nm). The open-pit mines to the north of the site bring the largest amount of particles by number and volume with peak number density near 120 nm and peak volume near 300 nm.

Aerosol flux measurements on the tower demonstrate substantial exchange of aerosols in both directions (into and out of the canopy) when averaged to 30-min fluxes. The net deposition over the nearly month-long measurement period is a small fraction of that variation. Deposition results agree with previous studies measuring aerosol deposition over forests in the < 1 μm size range. A local minimum of $v_d = 0.02$ cm s⁻¹ is observed at 80 nm, which is slightly less than the range suggested by the Emerson et al. (2020) parameterization, but not significantly different from this range (i.e., overlapping within the range of measurement uncertainty).

The local minimum of deposition velocity for sizes near 80 nm corresponds to the peak size of the number concentration PSD for smokestack emissions (80 nm) and is close to the peak size of the number concentration PSD for air from undeveloped forest areas (70 nm). This demonstrates the importance of correctly modeling deposition velocity in this range to accurately measure the number of particles depositing to forests. However, PSDs demonstrate that the bulk of mass (of sub-micron particles) is in the 150 to 400 nm range (or the 150 to 700 nm range for nearby open-pit mining emissions), so most of the aerosol mass deposited to the forest is likely due to impaction and interception by leaves and surfaces due to particle inertia.



Decoupling of the forest canopy is demonstrated at nighttime, with median lag times for concentration changes to be communicated from above the canopy to near the surface of up to 40 min. Median time lags during the day are between 2 and 5 min. The use of the flux/gradient method with the measured aerosol concentration gradient gives a size-integrated mass flux of PM1 which is between a factor of 2 and 9 smaller in magnitude than the flux measured by eddy covariance, depending on the parametrization used for the diffusion coefficient, $K$. The uncertainties in the averages determined by the flux/gradient method are comparable in magnitude to the averages (at the 68% C.I.), which demonstrates the substantial uncertainty in determining an average flux using the flux/gradient method.

Based on these results, it is recommended to use a modeling approach to investigate the relationship between time lags in the canopy and the modeled diffusion coefficients. This could help to determine if parameterizations of $K(z)$ can accurately reproduce the time lags seen in this and other studies and ensure that canopy decoupling can be accurately represented through diffusion-based modeling. This could potentially improve the agreement between the flux/gradient estimations and the eddy covariance measurements for PM1 at this location.

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



**Table 1: Average PM1 mass flux measured by eddy covariance compared to the average mass flux from 2-point flux/gradient measurements using different $K$ parameterizations. The confidence interval (C.I.) is given as the standard error of the mean. All units are ng m$^{-2}$ s$^{-1}$.**

| Method | $F$ | 68% C.I. |
|---|---|---|
| Eddy Covariance | −10.8 | 5.3 |
| Flux/Gradient (Eq. 2) | −5.7 | 6.3 |
| Flux/Gradient (Eq. 3) | −3.9 | 3.7 |
| Flux/Gradient (Eq. 4) | −1.2 | 2.5 |


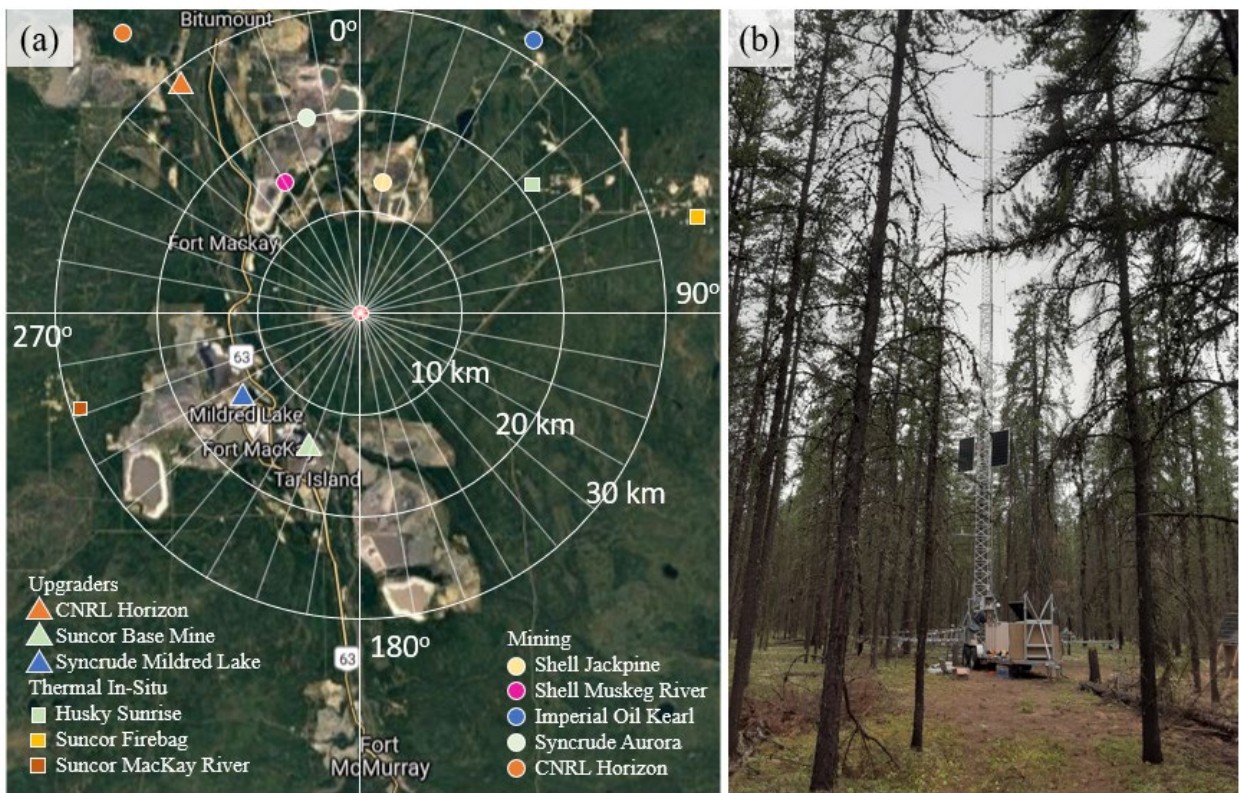

**Figure 1: (a) YAJP tower site location (red circle) and surrounding area with 10, 20, and 30 km radius circles and radial spokes at 10º increments. Image is © Google maps with upgrader and mine locations added from Davidson and Spink (2018). (b) The tower and the surrounding jack pine forest.**






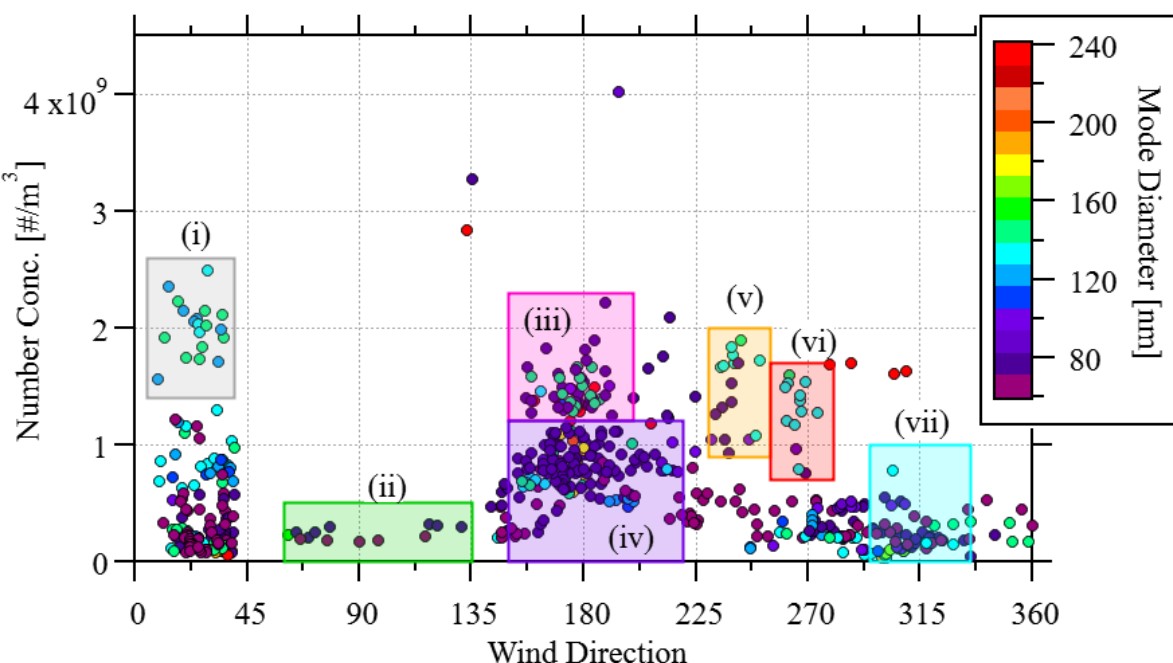

**Figure 2: Total particle number concentration (60 nm to 1 µm) with wind direction as 30-min averages. Markers are colored by**
**the peak or mode diameter of the number concentration PSD. To ensure consistent winds, only observations with less than 20°**
**change in wind direction in the preceding and following 30-min measurements are used. 7 sets of measurements in wind direction-**
**concentration space are identified for investigation.**



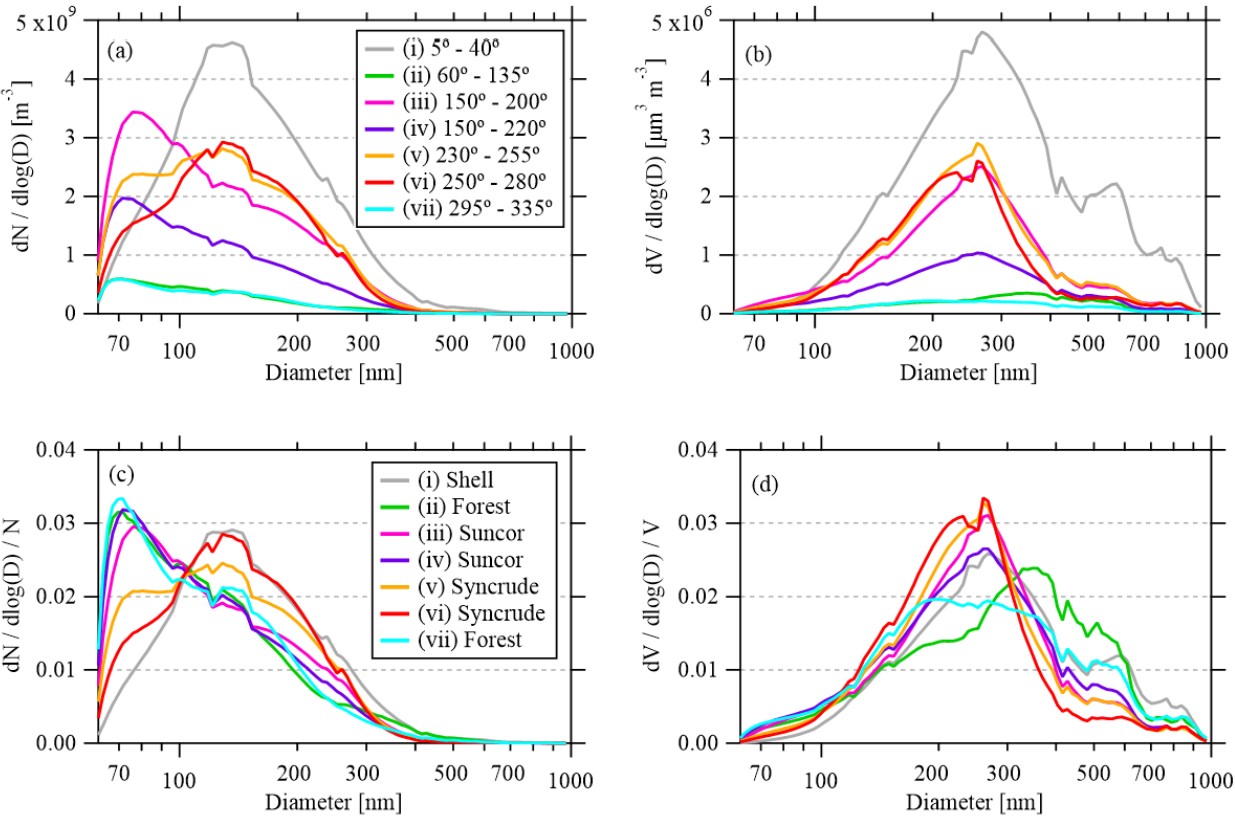

**Figure 3: Average size distributions for the sets identified in Fig. 2. Panels show (a) number distribution, (b) volume distribution,**
**(c) number distribution normalized by total number, and (d) volume distribution normalized by total volume. The identifiers**
**listed in the panel legend (c) are potential primary sources corresponding to the wind directions listed in panel legend (a).**





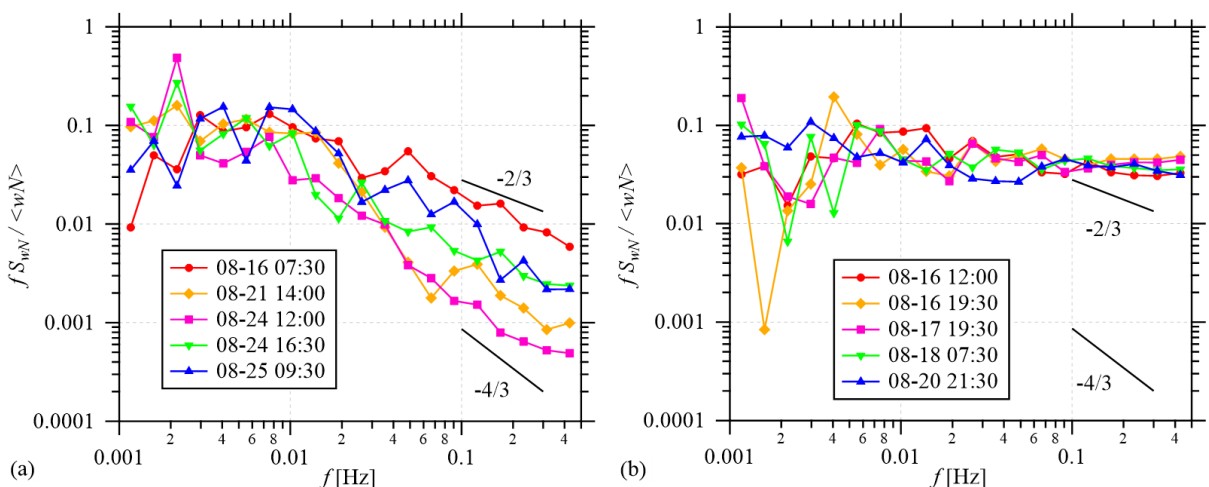

**Figure 4: Select aerosol number flux spectra through the measurement period demonstrating spectra with (a) an identifiable inertial subrange and (b) an unresolved inertial subrange dominated by noise. The power-law slopes of −2/3 and −4/3 are shown for comparison. The −4/3 slope is predicted by theory and the −2/3 slope is used for comparative purposes.**

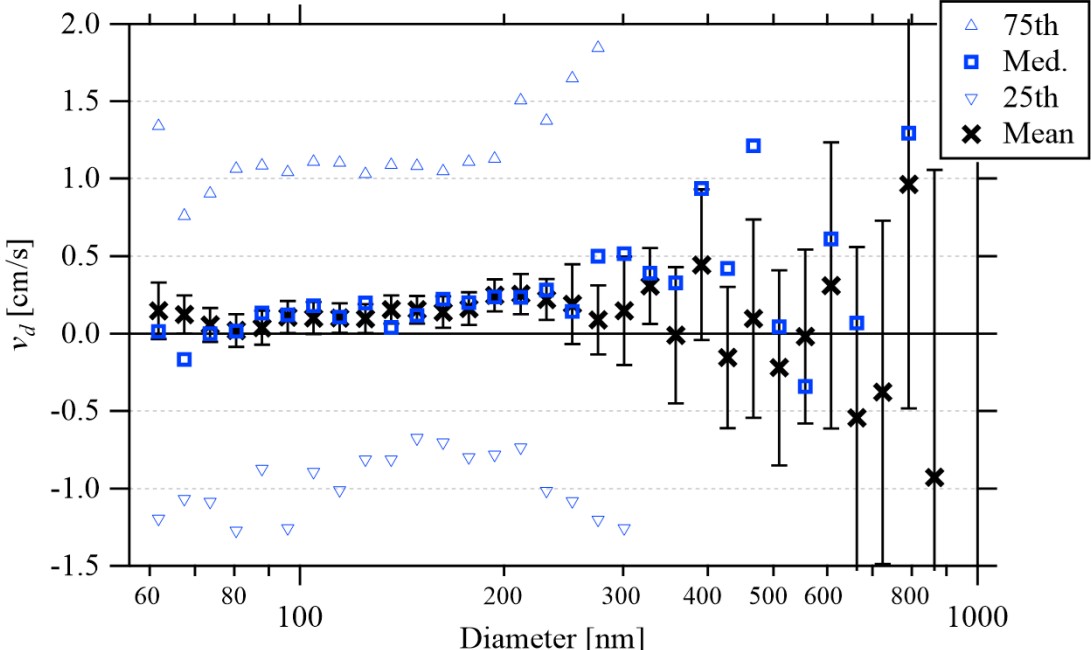

**Figure 5: Average and median deposition velocity with particle size. Error bars show standard errors of the mean (68% confidence intervals) and the triangles show the 25th and 75th percentiles. Percentiles for diameters > 300 nm are beyond the range of the graph as shown.**





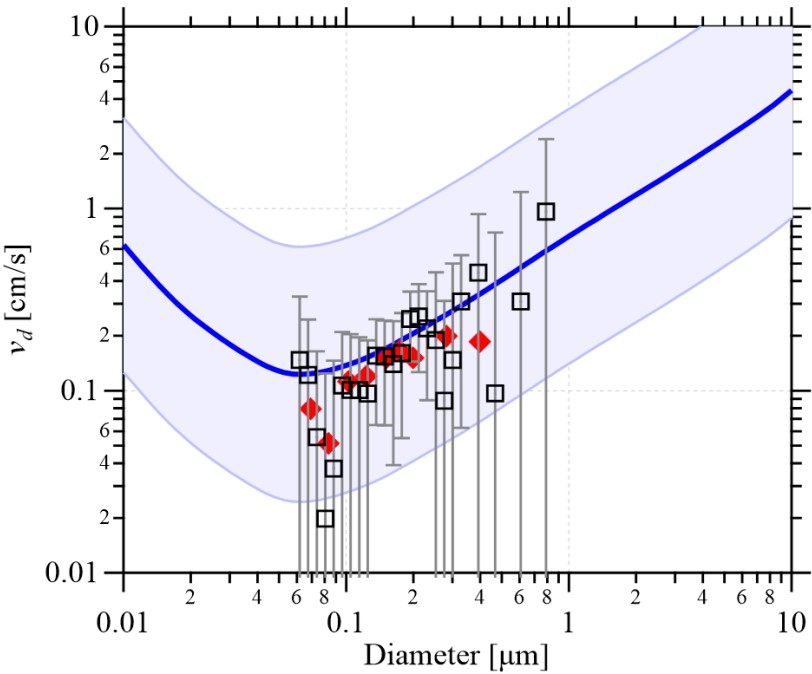

**Figure 6: Size resolved deposition velocities from this study (black squares) with error bars showing standard errors (68% confidence interval). Data are overlayed on the Emerson et al., 2020 measurements (red diamonds) and parameterization (blue line with 5× bounding range). Emerson et al. measurements are over a ponderosa pine forest and the parameterization is for needleleaf forest (their Fig. 1).**

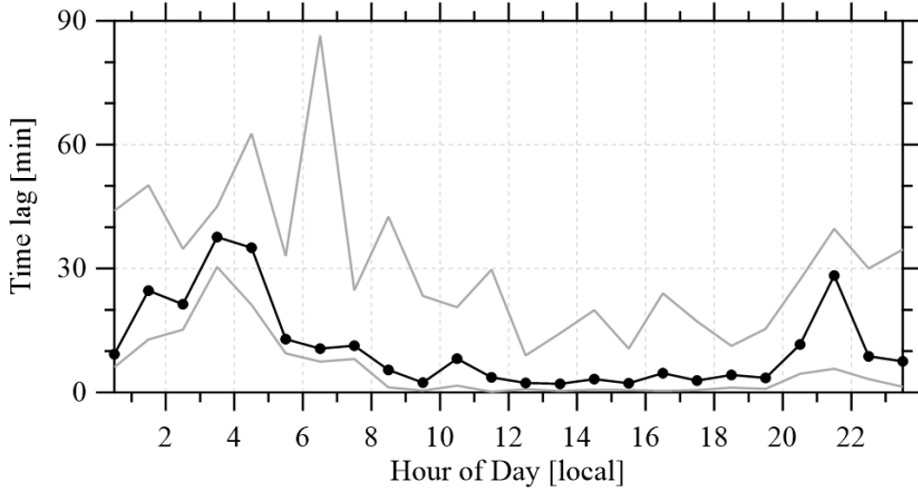

**Figure 7: Using the time series of TSI Dustraks the delay between changes in concentration at the canopy-top and identical changes near the surface can be determined and is here binned by hour of day. Medians, 10th, and 90th percentiles are shown.**