# Peer review of "Aerosol deposition to the boreal forest in the vicinity of the Alberta Oil Sands"

_Atmospheric Chemistry and Physics, 2022_

## Author Comment (AC1)

**Response to RC1: 'Comment on acp-2022-656', Anonymous Referee #1, 12 Nov 2022**

**General comments:**

The authors present a study of size-segregated particle size-distribution (SDP) and flux measurements at a forest site in the Athabasca Oil Sands Region of Alberta, Canada. The measurements enabled to correlate the SDPs to different particle sources in the Alberta Oil Sands. The particle flux measurement system enabled to determine the particle deposition rates in the size range from 60 nm to 1 mm. The particle flux measurements, in particular the size-segregated measurements, always impose a challenge due to high variability of flux sources and sinks and resulting high uncertainty in fluxes. The observed size-dependence of deposition velocities was in a good correspondence with latest parameterization and therefore the manuscript is a great contribution to experimental research on particle dry deposition.

Whereas there are no major concerns, a few topics would benefit from additional clarifications and potentially improvements. First, the particle number concentrations and SDPs exhibit systematic variation with wind direction. This is attributed to the downwind sources of pollutants or background concentrations in extensive forested areas. However, within the identified sectors (i to vii) there is significant variation and the authors have not attempted to find explanation of this variation in terms of atmospheric mixing (determined by stability) conditions. Simplest would be to differentiate the measurements into day-time and night-time, or into a few stability classes and test the hypothesis that the variation within segments is related to hour of day (perhaps also the source activity is dependent on time) or atmospheric conditions. A more sophisticated tool could be the source concentration footprint modelling, but presumable the assumptions of such models would be strongly violated (such as the assumption of surface sources) and as such not worth of considering. It is possible that the authors have made such analysis and neglected from the manuscript because of not finding strong evidence of the dependence on stability or time of day. In that case it deserves short explanation in the manuscript.

The deposition velocities are analysed as aggregates over all measurements (section 3.3). There is considerable scatter (and this is natural) and probably hard to differentiate more the measurements. However, attempt to separate day-night and/or some wind directions according to sectors i-vii (grouped to few subsets) might give additional insights and separate the conditions with different deposition/emission patterns, perhaps even to reduce the scatter when differently behaving samples are separated. Did you try this? If yes and this did not produce improvements, please mention in the manuscript.

AC1.G1: We thank the reviewer for this positive overview of the manuscript and for these excellent suggestions. We neglected to include analysis of diurnal variation in the PSDs. We have incorporated this analysis into the revised manuscript based on the reviewer's suggestion to separate the results based on time-of-day. The text of Section 3.1 (Source Characterization) is modified considerably, Figure 2 is changed to identify 5 sectors (from the original 7), and Figure 3 is replaced with new Figures 3 and 4, which segregate the results according to source sector and time-of-day.

We include the new text and figures below. The text rewrites lines 155 to 212 in the original submission. Although the text and figures are modified, the conclusions do not change. Small particles (~70 nm) are associated with stack emission sources and larger sizes (especially > 500 nm) are associated with nearby open-pit mining.

"The five sectors correspond to: the location of the Shell Jackpine site (0 to 40º), a forested area (60º to 135º), the Suncor upgrading facility and mines (140º to 225º), the Syncrude upgrading facility and mines (225º to 280º), and a second forested area (280º to 315⁰). The sectors are shown in Figure 1.

…

Particles size distributions (PSDs) for the sectors defined above are shown for particle number ($N$) in Figure 3 and volume ($V$) in Figure 4. Since the PSDs show a strong dependence on the time of day, we separate the observations from each sector into 12 PSDs, each comprising observations within a 2-hour period. Since oil sands mining and processing is a 24-hour operation for all facilities (Liggio et al., 2016), we assume the diurnal variation is due to meteorology and particle dynamics. The number PSDs show two strong peaks near 70 nm and 150 nm, which vary in relative magnitude by time-of-day and by sector. The volume PSDs (Fig. 4) have a primary peak near 250 nm and weaker, secondary peak near 600 nm. The time-of-day variation in the volume PSDs is more consistent between sectors than the time-of-day variation in the number PSDs. For the industry sources (Shell, Suncor, and Syncrude), higher peak values are seen through the day (08:00 to 20:00), which could be due to higher winds providing faster transport from the source to the measurement location. Average hourly wind speeds vary from 3.6 to 4.8 m s$^{-1}$ between 11:00 and 18:00 compared to 3.0 ± 0.2 m s$^{-1}$ outside those hours. The Shell sector shows the strongest secondary peak (~600 nm), which could be associated with the relative proximity of the Shell mines (~10 km) versus the Syncrude and Suncor mines (~15 km) as these larger particles may have deposited over the longer upwind fetch.

The number PSDs (Fig. 3) do not demonstrate a consistent day/night difference across the different sectors. While morning concentrations (08:00 to 10:00) are generally highest for the three industry sources, the mode diameter of the PSDs from the Shell and Syncrude sectors is near 150 nm, while from the Suncor sector it is near 70 nm. Peak number concentration for diameters near 70 nm suggests newly formed particles from upgrader stack emissions (Zhang et al., 2018)."

[Figure]

**Figure 1: Total particle number concentration, *N* (60 nm to 1 μm) with wind direction as 30-min averages. Markers are colored by the hour of day. To ensure consistent winds, only observations with less than 20º change in wind direction in the preceding and following 30-min measurements are used. 4 sets of measurements in wind direction-concentration space are identified for investigation.**

[Figure]

**Figure 2:** Number particle size distributions (PSDs) by time-of-day (averaged over 2-hours) for the four sectors identified in Fig. 2 (with the two forest sectors combined).

[Figure]

**Figure 4: Volume particle size distributions (PSDs) by time-of-day (averaged over 2-hours) for the four sectors identified in Fig. 2 (with the two forest sectors combined).**

A topic of a concern is the EC system frequency performance. It was explained that the size distributions were sampled at 1 Hz frequency (L. 102) and that the attenuation of signal at frequencies >1 Hz was corrected. It is not evident how the particle EC system frequency response was determined. Sampling rate is not equivalent to the frequency response of the system. In addition, in case of EC flux measurements it is important to determine what are the frequency response characteristics of the complete system consisting of the spectrometer and the rather long (32 m) sampling line. Please provide additional explanations to this experimental detail.

AC1.G2: This was another oversight. We returned to the lab to test the response of the system (with the original tubing length). The following text is added to Section 2.1:

"The instrument response time (with the 32-m tubing length) was determined in lab tests by measuring step-changes in concentration and fitting the response to a sigmoid curve (Horst, 1997). This gave a response time of $\tau = 0.9$ s. Petroff et al. (2018) determined a response time of 0.28 s for the UHSAS alone, suggesting that approximately 0.6 s of our measured response time is due to dissipation in the tubing."

Using a response time of 0.9 s instead of 1 s changes the average results by less than 1%. The deposition velocity figures (not Figs. 6 and 7) are modified for this new correction, but the difference is negligible.

The last part of the result, the particle mass flux inference from measured PM1 gradients raises the question on the applicability of the K-theory inside canopy. In general, the K-theory is poorly applicable inside the canopy and in case of more closed canopies might not be applicable at all (counter-gradient fluxes within canopy contradict the K-theory). This needs to be acknowledged in the manuscript. However, the forest at the study site was not closed and the reasons for large discrepancy are probably elsewhere. Within canopy deposition mostly occurs at the upper part of the canopy (in most canopies majority of leaf area is in the upper part of the canopy and deposition is more efficient at higher levels where more turbulence exists). Therefore, the average K evaluated for the height interval of the observations might be biased (the "resistance" equivalent to those K-parameterizations over estimated). Could this be partly responsible for discrepancy?

AC1.G3: We thank the reviewer for this excellent summary. We have attempted to incorporate these points into the manuscript. The modified text below is added after Eq. 3 (Eq. 2 in the original manuscript). Unmodified text is black.

"In cases of closed canopies, flux/gradient relationships are not generally applicable inside the canopy due to counter-gradient fluxes and modified stability within the canopy (Thomas and Foken, 2007). Measurements at this forest tower site outlined in Gordon et al. (2022) demonstrate good agreement ($R^2 = 0.83$) between $K$ determined by the measured flux and the gradient with a representative height of $z_m = 11$ m, which is roughly the middle of the canopy height ($h_c = 19$ m). This good agreement at this site may be due to the relative openness of the canopy (Fig. 1b)."

The relationship between the vertical variation of both $K$ and the deposition resistance is interesting but is very complex. LAI measurements are presented in our companion paper (Zhang et al., 2023). These demonstrate a peak in LAI density near a height of 4 m (due to smaller, new-growth trees), so it is unclear what a vertical profile of deposition would look like, especially given that the deposition resistance relates to both LAI and turbulence (as the reviewer points out). Based on these ideas, we add the following to the end of the discussion:

"These differences may be due to the oversimplification of the 2-point gradient approximation, which does not account for modification of in-canopy stability or counter-gradient fluxes. In addition to potential vertical variation in $K$, there may also be vertical variation in deposition resistance ($r = 1/v_d$) throughout the canopy. The 2-point gradient approximation only assumes deposition to the forest floor and not the tree and leaf surfaces. Any deposition which occurs within the canopy would likely reduce the concentration gradient and hence lead to an underestimation of $K$, which is here based on aerodynamic resistance only."

Added reference: Zhang, X., Gordon, M., Makar, P.A., Jiang, T, Davies, J., Tarasick, D.: Ozone in the boreal forest in the Alberta oil sands region, Atmos. Chem. Phys. Discuss., doi:10.5194/acp-2023-26, 2023.

Detailed comments

1. Line 30 and 32, remove repetition: "is key to correctly modelling atmospheric aerosol concentrations"

   AC1.D1: First instance removed.

2. 55, minimum located near 2 μm in diameter, presumably this was a local minimum at that large particle diameter?

   AC1.D2:  It is not a local minimum.  To clarify, we add "Both the Emerson et al. and Zhang et al. parameterization have a single minimum value over at 0.01 μm to 100 μm range."

3. 58, "free atmosphere", this was meant to mean the air layers above the forest? Free atmosphere denotes in meteorology the layer above the boundary layer within the troposphere.

   AC1.D3: We have replaced "free atmosphere" with "the air above the canopy".

4. 125, "three passes removed all high-frequency data", this is a bit bad wording, such processing had a purpose to remove unphysical spikes and not the high-frequency data, which should be retained for EC flux calculation.

   AC1.D4: "high-frequency data" is replaced with "data points".

5. 135 (and the concern above), how the frequency response 1 Hz was determined?

AC1.D5: See response to main comment (AC1.G2) above.

155, 201 (and the main comment above), did you try to differentiate the measurements according to hour of day or stability as an explaining factor?

AC1.D6: See response to main comment (AC1.G1) above.

6. 205, 206, should there first be sets (ii, iii, iv, vii), than set (i)? not all sectors are correctly assigned here, please revise.

AC1.D7: The text has been removed following the response to the main comment (AC1.G1) above.

7. 209, "ranging from 200 to 400 nm" is confusing, not ranging but being 200 and 400 for vii and ii, respectively.

AC1.D8: The text has been removed following the response to the main comment (AC1.G1) above.

8. 236, the instrumental noise, if not correlated with wind measurements, should not appear in co-spectra. But the fact that signal to noise ratio is small affects determination of good co-spectral shapes.

AC1.D9:  We modify this text to "…possibly due to a lower signal-to-noise ratio, which can affect the co-spectral shape."

9. 239, How diffusion in the sampling line would affect the flux? By affecting the frequency response of the system? What physical process is meant here, diffusion of aerosol particles to sampling line walls (this should not be significant for larger than 60 nm particles)? Or did you mean that the laminar flow (not sure what was the flow regime) caused high-frequency damping?

AC1.D10:  We meant the former and have change the text to "diffusion in the flow direction within the sampling tube, leading to a reduced system frequency response."

10. 245-248, might be matter of taste, but "strong spectra", "strong/low flux measurements" seem loose wordings which might be better to replace.

AC1.D11:  We have modified the wording of this paragraph as follows (black is original text, blue is modified): "Approximately 83% of the spectra with $S < -2/3$ are measured during the daytime (between 07:00 and 17:00) when there is greater flux of aerosols into the canopy (due to the increase in turbulent mixing during the day). This demonstrates that the presence of the inertial subrange is associated with fluxes that are greater in magnitude than the fluxes associated with flat inertial subranges ($S \sim 0$). Although the lack of an inertial subrange may indicate a significant noise to signal ratio, all the spectra (including those associated with lower flux and concentration values) are

included in our analysis since removal of these data would introduce a daytime bias in the results."

11. 304, which stability function was used? The function applicable to the atmospheric surface layer? I doubt that it would work inside the canopy.

AC1.D12: We have added "The stability parameter ($\phi$) is determined from the Obukhov length ($L$) following Garratt (1994) as

$$\phi = \begin{cases} \left(1 - 16(z/L)\right)^{-1/4} & -5 < z/L < 0 \\ 1 + 5(z/L) & 0 < z/L < 1 \end{cases}, \tag{3}"$$

As discussed in the response AC1.G3 above, the point of this exercise was to investigate how well this simplistic approach would work and we have added text (outlined in AC1.G3) to try and make that clear and to add "modified stability within the canopy" as one of the reasons explaining the discrepancies shown in Table 1.

12. 500. Add to the caption of Fig 5 what is N (the aerosol number concentration for sizes between 60 nm and 1 µm).

AC1.D13: To be consistent here, we have added N to the y-axis of Fig. 2 and added the variable to the figure caption. In Fig. 4 (as suggested), we add the definition of N (with size range) and $w$.
* * *
**Response to RC2: 'Comment on acp-2022-656', Bruce Hicks, 09 Dec 2023**

I am a fan of the field research activities of the teams at York and Guelph Universities. Working with the Environment Canada micrometeorology group, they have provided decades of revealing results regarding air-surface exchange, primarily involving forests. The present submission continues the progress. The experiments described were based on new sensors that permit extension of particle covariances into size ranges considerably smaller than previous studies.

AC2.1: We thank the reviewer for this very encouraging comment and summary.

Line 18. Please explain "PM1." Later on, we find PM2.5 PM4, PM10. Maybe define each at the outset.

AC2.2: We remove the abbreviation PM from the abstract and instead use "aerosol particles (with diameters < 1 µm)" and "Aerosol mass fluxes (diameters < 1 µm)". In the methods section (where the DustTrak instrument is introduced), we add the text "These measurements are size resolved into total mass for diameters less than 1 µm (PM1), 2.5 µm (PM2.5), 4 µm (PM4), and 10 µm (PM10).".

Line 28. My understanding is that the *de minimis* aspect of particle health effects remains contentious. I recommend softening this statement (attributed to Kappos et al., 2004).

AC2.3: We have changed the text to "…with some studies showing health effects even at very low concentration exposures".

Line 32. Delete sentence. It repeats what has already been said.

AC2.4: Sentence deleted.

Line 45. To help clarify the presentation, the terms "Aiken" and "accumulation size" might best be introduced earlier. The text so far has concentration on numerical sizing. These new terms are introduced without explanation.

AC2.5: Since this is the only reference to Aitken and accumulation modes in the manuscript, we have chosen to replace this with diameter sizes (from Emerson et al.) of < 500 nm and > 2 μm.

Line 48 et seq. I am quite unimpressed by the Finland work and wonder whether is appropriate to think of it as providing "experimental evidence." It seems to me that the major contributor to their conclusions is the model they use.

AC2.6: While we agree that the results of this study may be influenced by the choice of model, we which to include this demonstration of the "well" over a forest, since eddy-covariance flux measurements in this region are limited. However, we have softened the language in the text and have removed the line which gives the deposition velocity values.

Line 53. Watch the font change. Here, and elsewhere.

AC2.7: Our pdf version does not show a font change here and we have double-checked the manuscript to ensure it is a consistent font. Hopefully this is not a file conversion issue and this will be resolved with proofing.

Line 54. OK. I cannot resist. As far as I am concerned, forests differ considerably from other vegetated surfaces, in that the subcanopy air space of a forest serves at a constrained chemical reactor is which all sorts of particle generation and growth processes flourish. The "deposition velocity" measured above the canopy is then the net consequence of an upflux of particles of recent origin and a downflux of aerosols from somewhere upwind. All else follows. But I like the 70 nm minimum point. My own data indicate about 100 nm, but I figure this depends on the site and its surroundings and so I do not look for generalities.

AC2.8: We agree this is a very complex process and we keep this in mind for future studies and analysis.

Line 70. I suggest that it would be a good idea to squeeze this description of the chemical composition of the aerosols into somewhere earlier, so that people (like me) who automatically think in terms of pinenes and the like do not head down an inappropriate path.

AC2.9: We agree this makes more sense closer to the discussion of the AOSR (before deposition is discussed in more detail). We have moved the text that begins with "Several aircraft-based studies…" up to the 3rd paragraph in the Introduction.

Line 103.  At this point, I started reading some of the basic aerosol agglomeration/growth literature (my favourite Friedlander's "Smoke, Dust and Haze"). After recognizing that I no longer understand much of what seems relevant, it occurred that you guys must have done the relevant tests.  My main concern is that the sample size spectrum at one end of the 32 m tube will differ from that at the other, due to particle processes occurring during transit.  If there is a change, how do you account for it?

AC2.10: Laboratory experiments with the UHSAS (including the 32 m tubing length) with sized particles (from 100 to 500 µm) demonstrated no significant line losses and no correlation of line loss with particle size. While there is a 9-s residence (delay) time through the line, there should be no significant particles processes occurring during that time that wouldn't occur in the atmosphere before sampling.

Line 130.  I always wince when low-speed observations are rejected. To my mind, these are the most variable and hence the best to focus on.  There is also a tacit assumption involved – that the transport of particles is somehow associated with the flux of momentum. This is not the right time to look at this in detail, but if you have other covariances (e.g. c'T', c'u') and especially the partial correlations that arise, then a little exploration could be entertaining.

AC2.11: We agree this further exploration would be very interesting but may be beyond the scope of this study.  We refer to the reviewer's point below (where no change to the present text is recommended) and we hope that achieving of the data (and our own follow up analysis) might lead to further investigation using these measurements.

Line 153.  This seems to say that "While HYSPLIT could work better, in reality it doesn't."  No surprise here.  No local model constructed using mesoscale outputs can improve on what local eyeballs report.

AC2.12: While HYSPLIT can be very useful, at this scale and in this region, it is our experience that local wind measurements better demonstrate source determination.

Line 236 – 251.  I think that this discussion illustrates the complexities and insecurities of extending well verified flat-earth and conventional meteorological flux experiences to issues of local (and very practical) importance.  The discussion is along the lines advocated by micrometeorological purists of the Obukhov community, but to my mind the reference to power-law slopes of -1, -7/3, -4/3, -2/3 is sufficient for me to prefer a different approach, based on the confidence with which measured particle covariances represent the statistical distributions that are expected and as measured.  To this end, I would prefer to look at the details of the , and analyses, and to examine these with consideration to the relevant correlation coefficients (mainly partials) derived from similar measurements of and . This has been very informative in the past, but flies in the face of what micrometeorological convention and its perfect-site

advocates recommend. To my mind, the essence of air-surface exchange is statistics, and examination of the statistics is the only way to address many of the issues that arise.

I do not recommend changing any of the text now presented, but I request that the authors consider my views and examine how their data archiving could provide a more statistically satisfying quantification of uncertainties. I suspect that there is a goldmine of relevant data ready to be mined.

AC2.13: We will provide achieved data and also hope to continue this analysis ourselves.

Section 3.3. A couple of things concern me. In particular, Figure 6 appears to be of averages and standard deviations computed arithmetically but plotted on a logarithmic scale. Why? On first principles, the individual quantifications of $V_d$ are ratios of covariances to averages, both quantities being subject to large statistical uncertainty. The distribution of $V_d$ should then be log-normal (or close to it), and the plots of Figure 6 should be of the appropriately transformed data (geometric means and relevant error bounds).

AC2.14: This is an interesting point and we thank the reviewer for thinking of this. This was graphed on a log scale following the convention in many previous studies. The log scale in these review studies (including Emerson et al. 2020, where this data is from) makes sense given the wide distribution of scales. But given the small range of values presented here a linear scale seems more appropriate. The reviewer's comment motivated us to look at the distribution of $v_d$ and we can confirm that it follows a normal distribution (not log-normal). We add text in the 1$^{st}$ paragraph of Section 3.3 as "A standard error based on the variance is used here based on the normal distribution of the measured $v_d$ values (not shown).". Hence, we have modified Fig. 7 (Fig. 6 in the original submission) to be linear on the y-axis, which also allows us to demonstrate the negative values.

Section 3.4. This is a very welcome discussion. It is based on familiar flat-earth time-stationary Fickian stuff, that I have never accepted as appropriate for any sub-canopy environment. I like the results presented in Table 1 and thank the authors for going through this exercise. My interpretation of Table 1 is that none of the gradient-interpretation analyses yields results statistically different from zero, and hence none gets close to what eddy covariance indicates. However, I am nervous about the tabulation. I suspect that a different conclusion could be drawn if the statistics were based on log-transformed results.

AC2.15: As discussed above (and added to the text), the $v_d$ values are normally distributed (not log-normal), so this comparison (with confidence intervals) should still be logically sound.

Lines 353 - 358. Careful. Compare what is said here with the my interpretation of Table 1 above.

AC2.16: Since we have demonstrated the normal distribution of $v_d$, we believe these interpretations of the confidence intervals are correct.

Lines 358 – 363. There is something unsatisfying about using a model to determine if another model needs to be changed. My opinion is that the analysis yielding Table 1 has already shown that the sub-canopy use of conventional diffusivity relationships is not highly profitable. I have yet to find a counter example.

AC2.17: This was poorly worded. We rewrite this as "… it is recommended to use a more detailed modeling approach, such as the high-resolution, 1-dimensional canopy model outlined in Zhang et al. (2023), to investigate the relationship …". While the use of a simple approximation outlined in Section 3.4 demonstrates the limitations of this approach, a more detailed high-resolution model investigation may be able to tell use more about mixing within the canopy.

Added reference: Zhang, X., Gordon, M., Makar, P.A., Jiang, T, Davies, J., Tarasick, D.: Ozone in the boreal forest in the Alberta oil sands region, Atmos. Chem. Phys. Discuss., doi:10.5194/acp-2023-26, 2023.

Figure 7. Of the three lines plotted, only one is detectable.

AC2.18: We have changed the grey lines to more clearly visible black lines.